# Enhancing Maritime Crew Resource Management Training by Applying Resilience Engineering: A Case Study of the Bachelor Maritime Officer Training Programme in Rotterdam

**Jaco Griffioen [1], Monique van der Drift [1,\*] and Hans van den Broek [2]**

1  Rotterdam Mainport Institute, Lloydstraat 300, 3024 EA Rotterdam, The Netherlands; j.r.griffioen@hr.nl
2  Research Centre for Sustainable Port Cities, Heijplaatstraat 23, 3089 JB Rotterdam, The Netherlands; j.van.den.broek@hr.nl
\*  Correspondence: m.van.der.drift@hr.nl

**Abstract:** This paper sets out to enhance current Maritime Crew Resource Management (MCRM) training, and with that to improve the training of technical and non-technical skills given to bachelor maritime officers. The rationale for CRM training is improving safety performance by reducing accidents caused by human error. The central notion of CRM training is that applying good resource management principles during day-to-day operations will lead to a beneficial change in attitudes and behaviour regarding safety. This article therefore indicates that enhanced MCRM should play a more structural role in the training of student officers. However, the key question is: what are the required changes in attitude and behaviour that will create sufficient adaptability to improve safety performance? To provide an answer, we introduce the Resilience Engineering (RE) theory. From an RE point of view, we elaborate on the relation between team adaptability and safety performance, operationalized as a competence profile. In addition, a case study of the 'Rotterdam Approach' will be presented, in which the MCRM training design has been enhanced with RE, with the objective to train team adaptability skills for improved safety performance.

**Keywords:** maritime crew resource management; resilience engineering; adaptability; training; maritime safety





## 1. Introduction

The aviation industry has been utilizing Crew Resource Management (CRM) training for more than three decades, which extends from cockpit crews to other groups within airlines. The objective of CRM training is to effectively use all available resources, i.e., people, equipment, and information in order to manage error. However, it has been realized that CRM is not a panacea to eliminate error and assure complete safety [1]. The current evidence for the effectiveness of CRM training programs is not perfect due to a lack of systematic studies, and the effect on safety is still unclear; nevertheless, people have become to realise that CRM training produces a desired change in attitude and behaviour [2].

The maritime industry introduced CRM training for bridge officers a few years after civil aviation did so for cockpit crews. Despite this adoption, the evolution of the training is lagging in respect to standards of competence, assessment criteria, the duration and frequency of refresher training courses, and the extension of the training to other parties, i.e., non-bridge crews, in the maritime domain. Since the revision of the training requirements for seafarers in 2010 by the International Maritime Organisation (IMO) [3], the CRM course became mandatory for ship officers and Maritime Training Institutes had to implement this training in their curricula. However, the IMO does not prescribe how CRM-specific learning objectives can be trained as effectively as possible. An exploratory questionnaire, draw-up by the authors of this paper, has been sent out to lecturers from nautical institutes worldwide, and shows that every institute applies CRM training differently. In addition, studies have indicated that the effectiveness of the CRM course is low or undetermined [4].

The aim of this paper is to provide suggestions on how to enhance MCRM training. First, from a training design point of view, CRM is suboptimal, i.e., training takes place on an incidental basis and training is scheduled in different years of the curriculum. Therefore, training structure, frequency, and integration within the curricula should be imposed. Moreover, a better transfer from classroom to everyday practice should be enforced so that the effective use of training outcome will be improved. The second aim of this paper is to understand maritime safety from a Resilience Engineering (RE) point of view. According to RE, safety can be achieved when a ship has sufficient adaptability, i.e., resilience, to cope with a large range of operational dynamics [5]. Although literature indicates that about 80% of accidents are caused by human errors [6], this denies the fact that crews are part of a socio-technical system and therefore have to rely on the relationship between people, technology, the organization and the environment for overall safety performance [7]. Therefore, viewing humans as a single source of failure is based on an outdated linear and deterministic approach on safety [8]. Moreover, it denies the fact that humans turn difficult or near accidents around, preventing accidents and disaster from happening. Statistically speaking, accidents occur in 1 out of 10,000 of the total ship operations [9]. Which also means that one should expect things to go right 9999 times out of 10,000. Thus, operating a ship is complex as multiple interdependencies exist. We know from experience that a ship should be seen as being part of a larger traffic and transport system, which contains many different actors with different roles and responsibilities. For instance, the team consists of the individual crew members that make up the team and, when entering the port, a team needs to deal with other parties such as the pilot, linesmen, operators of tugboats, and vessel traffic centre operators. For the day-to-day operation, a team (captain and crew) on board must also deal with a shore organization, all of which contributes to the complexity and dynamics for which adaptability is required at almost every segment of the voyage. It is important to realise is that a ship's crew consisting of only individuals without cohesion and interaction will not be able to cope with challenging situations because no information will be exchanged and there will be no clarity about the division of work and responsibilities. Therefore, team adaptability, cohesion, and interaction with all the actors within the system, in expected and unexpected situations, should be enforced. From an RE perspective, a competent team is a prerequisite for the prevention of accidents and incidents [7]. Therefore, the key question that is addressed in this paper is: what are the required changes in attitude and behaviour that will create sufficient adaptability to improve safety performance? We have presented RE theory as a framework for connecting CRM training objectives and increased safety performance.

The 'Rotterdam Approach' will be presented in which the CRM training-design is enhanced with RE principles. Currently we do not have any evidence that the 'Rotterdam Model' will contribute to safety in general, nor is it that the aim of this study. What we are aiming for, however, is to provide suggestions on how to enhance MCRM training and how to structure the training program. We are open to feedback and would like to invite everyone to share experiences in order to learn from each other and improve the effectiveness of the MCRM training.

The article is structured as follows: Section 2 provides a brief discussion of the development and implementation of CRM training in maritime education and training. Section 3 offers an RE approach for safety in maritime operations. Section 4 describes the enhanced MCRM training design by adding RE and effective CRM training elements to better match the CRM objectives, creating team adaptability for safety performance. In Section 5, MCRM training design practice enhanced with RE will be outlined and specified as the Rotterdam approach, followed by a summary and discussion.

## 2. Maritime Crew Resource Management (MCRM) Training

Since the major revision of the International Convention on Standards of Training, Certification and Watchkeeping for Seafarers (STCW) in 2010, Crew Resource Management (CRM) training has become a mandatory part of the training program for deck and engine

room department officers [3]. CRM training is not new in the maritime industry and was at most times referred to as Bridge Team Management (BTM), Maritime Crew Resource Management (MCRM), Maritime Resource Management (MRM), Bridge Resource Management (BRM), and Engine room Resource Management (ERM). The courses were initially carried out on a voluntary basis with a duration that varied between two and five days.

The origins of CRM can be traced back to a series of accidents that occurred in aviation in the 1970s [1], where research showed that most of the accidents were caused by failures of interpersonal communications, decision making, and leadership. A new type of training program was developed to reduce "pilot error" and was initially called "Cockpit Resource Management" and later renamed "Crew Resource Management". In parallel, the first-generation CRM training for the maritime industry was developed in response to several maritime accidents in the mid-1970s on vessels with well-trained crews, where bridge teams, including the pilot, did not work as a team to support each other [10]. These simulator-based training courses were initially introduced to primarily train skills of passage planning and to enable an understanding of how to organise the duties and effective communications given by a bridge team to plan and execute the voyage [11] and is often named as Bridge Team Management (BTM). This course is still available, with the focus on complying with existing means of navigation procedures. In the beginning of the 1990s, a group of eight organisations [12] converted the airline industry's Cockpit Resource Management course to a course for the shipping industry named "Bridge Resource Management" (BRM).

In contrast with the Bridge Team Management (BTM) course, where the focus is on the technical skills, the objectives of the Bridge Resource Management (BRM) course is to change attitudes and behaviour to apply good resource-management practices during everyday operations, also known as non-technical skills or soft skills. The goal of both the BTM and BRM training programs is, in general, improving safety performance by reducing accidents caused by human error. Research on the effectiveness of CRM trainings has not found a clear link between the training and a reduction of accidents [2,13,14], although there is some evidence of safety improvement in maritime transport. It is difficult to say whether this improvement is specifically due to the CRM training or a more general acceptance of procedures on board ships [15]. By sending officers to a training course, the expectation was that erroneous behaviour could be unlearned and replaced with more appropriate behaviour. However, attending such a course does not in itself make a ship safe, and the following factors must be taken into consideration:

- Bridge officers are part of a socio-technical system, where they must also deal with the organisation culture [16]. In the opinion of the authors, it is for that reason worthwhile or even essential to train all the parties involved and not just the officers, but the complete crew and all people in shore organisations who have an influence on safety at sea and the work on board a ship. This was already recognised in commercial aviation in the 1990s, where CRM began to be extended to other groups within airlines such as flight attendants, dispatchers, and maintenance personnel [1]. In the same way, the maritime industry also realised that not all accidents originate from the bridge, and the training course was renamed to Maritime Resource Management (MRM), with additional target groups such as engineers and shore-based personnel [12].

- The training design and implementation of the courses were not standardized. The goal of the STCW has been to establish a common, international training standard for seafarers from various nations, which mentions the minimum standard of competencies for each rank in its tables [3]. The tables describe the required knowledge, understanding and proficiency and the way these can be assessed, but are still open for more than one explanation. In order to implement this in a better way, and to achieve the standardization of the content that is taught across countries, the IMO publishes model courses. However, these IMO model courses are not mandatory, and countries can make their own design and implementation choices if these fulfil the minimum requirements of STCW. From their study on the effects of classroom BRM training [4]

and their reference to many years of CRM research and practice, recommendations for BRM training design and implementation were made, which are used in this article to develop an improved competence-based CRM training design and is described in Section 4.

- Evaluation of the change of behaviour during the training and the transfer of the behaviour on board. In order to monitor and assess the effective behaviours, behavioural markers are used. Behavioural markers are defined as "observable, non-technical behaviours that contribute to superior or substandard performance within a work environment" [17]. They are usually structured into a set of categories (e.g., decision-making and situational awareness). Normally, these categories are then subdivided into more specific nontechnical skills or elements.

- In Gatfield's research [18] on behavioural markers for the assessment of competence in crisis management, he states: 'For a behavioural marker to be an effective assessment metric, it needs to be relevant to the competence being assessed. It should be easily evaluated as a demonstration of good or poor behaviour, easily observed and should occur quite frequently (p. 118)'. In aviation industry, there are some standards available, e.g., the University of Texas (UT) system and the NOTECHS scheme, but there are many more behavioural-marker systems in existence. In the maritime industry there is no standard available yet, but there are some examples from research papers [16,19,20] which give a good start to developing a standard. It is important that the training is followed up on board, using information from team members.

- Duration and frequency of refresher training courses. The duration of the training is limited because officers attend the courses during their leave period. It can be questioned if a change of behaviour can be realised in such a short period, and if so, what the results are over a longer period. Research from military aviation shows a 25 percent decline in attitude towards CRM after 12–14 months [21] and therefore refresher training courses are scheduled every three years [22]. Although there is no research on retention of CRM skills in the maritime industry, CRM training should not be a single experience [1]. In our opinion, all necessary competences need to be trained in a continuous program; moreover, during the career on board, refresher trainings are critical for long-lived results. The aviation industry has a recurrent training program where CRM competences are trained every three years, whereas the STCW only mandates refresher courses for safety and medical trainings, where the CRM is not included. However, there are shipping companies who refresh the technical and non-technical skills on a regular basis in simulator centres with the teams that work on board. Moreover, officer and crew conferences are used to discuss these non-technical skills, but this does not give the opportunity to observe the behaviour as in simulator centres.

From the above, it can be concluded that the current CRM trainings are still immature, and improvements must be made in order to enhance the safety culture in all the socio-technical components of the maritime industry.

As of 1 January 2017, administrations must comply with the Manila amendments to the STCW Convention Code, in which several changes were implemented to, among others, Bridge Resource Management, Engine Room Resource Management, and Teamwork and Leadership courses at both the operational and management level. In order to comply with this requirement, the maritime training institutions had to implement these subjects in their curricula. The STCW requirements are listed in the following tables [3]:

Table A-II/1: Bridge resource management and application of leadership and teamworking skills

Table A-II/2: Use of leadership and managerial skills

Table A-III/1: Engine-room resource management and application of leadership and teamworking skills

Table A-III/2: Use of leadership and managerial skills

The learning objectives for these courses are:

- allocation, assignment, and prioritization of resources,
- effective communication,
- assertiveness and leadership,
- obtaining and maintaining situational awareness,
- consideration of team experience including workload management.

As student officers are at school for multiple years, a better option is to schedule training throughout the school period, which is in line with the recommendations of the design and implementation of CRM training [4]. The nautical school in Rotterdam, part of Rotterdam Mainport Institute (RMI), has scheduled the CRM as a four-year training program and has integrated this with the technical skills. The program is described in detail in Section 5.

## 3. Resilience Engineering Approach for Safety in Maritime Operations

Chapter two shows that the goal of CRM training programs is, in general, about improving safety performance by reducing accidents caused by human error by training non-technical skills. However, the problem with this traditional approach, defined by Hollnagel [9] as Safety-I, is that most accidents cannot be explained by reducing complexity to one underlying factor, e.g., human and organizational errors, as was the case in some accidents in the past such as the Titanic, Harald of Free Enterprise, and the Costa Concordia [23,24]. Instead, accidents originate as a complex phenomenon from the daily operational variability of a socio-technical system [25]. The variability, according to Hollnagel [5], is a wide range of operational circumstances and conditions coming from the environment (exogenous variability) and from the inside (endogenous variability) of the subsystems of a socio-technical system, such as individuals and/or groups and/or technical systems.

RE came into existence as a reaction to the linear and deterministic way of thinking about safety by stating that accidents relate to a failure of the socio-technical system [8]. Hence, RE focuses on the interaction within the socio-technical system, aiming at performing effectively in everyday conditions; in other words, to perform everyday work successfully [26]. According to RE, the human factor is no longer the cause of errors but makes a positive contribution to the safety of such systems, because people, with their intelligence and creativity, can adapt to current changes and to shortcomings in the system design and unplanned (emergency) situations [27]. Hollnagel uses the term Safety II to distinguish this new paradigm of safety.

RE is a young field of science aimed at a new safety management paradigm of increasing the adaptability of socio-technical systems regarding disturbances [28]. The term adaptability has its origin in nature, where ecological systems can be resilient if they are able to undergo changes while preserving the existence of their function. The preservation and efficiency of the organization is important for the resilience of an organization [29]. In the early years of the 21st century, safety in high-risk domains was added because these domains are well known for their adaptability to complexity and high-risk technologies [30]. System goals, such as efficiency and safety, often require trade-offs because they cannot always be attained simultaneously. Consequently, human operators in high-risk industries are often forced to improvise by adjusting tasks and to discover flexibility by creating workarounds in order to be able to cope with limited resources [31]. The adaptability in high-risk domains, highlights how systems successfully adapt their behaviour to the shifting demands in the environment to stay in control and produce a stable performance output [32]. The shipping industry can be identified as a high-risk domain due to a significant complexity of operation, being part of a socio-technical system and the continuous adaptation to a changing environment.

This complexity of the shipping industry, not only caused by conditional changes such as weather, current, and wind but also caused by changes in the socio-technical system due to the economic pressure of an industry characterised by "faster, better and cheaper" [33], is a major driver of continuous adaption and change. As an illustration of making the

shipping industry better, the shipping industry is currently undergoing substantial changes through an increased complexity in technology as a solution to the human error. Despite technological solutions, accidents still occur. According to RE, in these accidents, the system is not resilient to unexpected situations. It is also noted that the human element of the maritime system has not yet evolved in the same way as technology has developed, not paying attention to the capabilities and limitations of the human factor [34]. The technology solutions also result in creating new paths to failure and new demands on workers [27] and ensure that the seafarer is being pushed further away from the original core processes, such as navigating or operating machinery, towards a more supervisory role [35]. As a result, non-technical skills are becoming increasingly more critical for adaptability in this complex world, where the seafarer has a supervisory role.

In accordance with RE, the added value of CRM training is to increase the ability of teams to deal with a wide range of operational circumstances and conditions. Non-technical skills play a major role in team adaptability, which is needed in order to deal with a wide range of operational variabilities, e.g., to adjust team members' responsibilities and tasks, adapting to procedures, (de)briefing information, creating common vision and goals and promoting confidence in the team. From this, it follows that effectiveness of CRM training, or non-technical skills training programs in general, cannot be measured directly in terms of accident reduction, but instead, should be measured by the adaptive power of the teams.

This new understanding of adaptability is relevant for improving CRM training design. The goal of CRM training programs, from an RE perspective, is improving safety performance by focusing on the adaptability of a team that is part of the socio-technical system in expected and unexpected situations by training non-technical skills. As defined by Hollnagel [36], resilience is 'the ability of a system or an organisation to react to and recover from disturbances at an early stage, with minimal effect on the dynamic stability (p. 16)'.

The focus of CRM training from an RE point of view is related to the meaning of adaptability, of which, according to Hollnagel, has two challenges. The first challenge is that adaptability is more than just the adaptability of the individual. A team, more than individuals, has a broader repertoire of capacities, experiences, and networks to fall back on in the "challenge" of adapting processes and performance to the requirements of the situation [37]. Teams and especially team collaboration are crucial links which allow organizations to be assertive prior to, during, or following changes and disturbances, in order to improve adaptability and be resilient as team [27]. Complexity, as a second challenge, focuses on all the interactions in this socio-technical system a team must deal with, both in expected and unexpected situations where, according to Cilliers [38], a system cannot be fully described nor fully controlled, as it is harder to predict what the outcome will be.

For improving CRM training design, further operationalization of RE needs to be implemented. The different stages of adapting prior to, during, or following changes are distinguished for an RE approach to CRM training. In order to be able to analyse the stages of everyday work and adaptability during changes in socio-technical systems, Hollnagel and Woods [27] defined four critical resilience abilities to interact and coordinate during these changes:

- Responding, knowing what to do.
- Monitoring, knowing what to look for.
- Anticipating, knowing what to expect.
- Learning, knowing what has happened.

The abilities are complex due to the many different operational variabilities, i.e., situations, expected and unexpected circumstances, developments, and opportunities. In expected situations, for instance, monitoring the actual day-to-day operations requires the ability of searching for input received from both the inside and the outside of the system, i.e., coming from tasks, information, and actors in the network [39]. Responding requires

the ability to address the actual task. Anticipating requires the ability to plan and share a forecast of the coming voyage, ensuring that people know what to look out for and what to expect in order to be able to respond.

In unexpected/emergency situations, to monitor a threat, team members must have the ability to integrate, synthesize, and share information in order to know what to look for, as tasks demand a shift in unexpected situations. During these unexpected circumstances, complexity creates gaps in the procedures of daily operations where the team functions. Some authors have suggested that "adaptive coordination" as a central requirement in a critical situation must use idle times for information sharing rather than responding immediately. They suggest anticipating by looking for information, in order to know what to expect, and dynamic redistribution of workloads among team members is needed in order to know what to do [6].

In order to learn, it is important to know what significant operational changes are and to know how a team adapts to stressful situations. During on board briefings and debriefings, the focus should be on two principles of RE [9]: first, it is important to find out "why daily operations go right ", as safety, according to Hollnagel, arises when as many day-to-day operations as possible go right in order to learn from the adaptability of the team. In order to be able to address everyday work, one should ask one another to "explain the situation as you see or saw it" [40]. In this way, it is possible to find out "why daily operations go right", rather than asking "why do or did you do this" in order to find out why operations go wrong. Second, it is important to focus on "Work-As Done" (WAD) as a baseline of everyday work, as safety arises from understanding WAD. Narrowing the difference between "Work-As-Done" and "Work-As-Imagined" (WAI) creates a better understanding of the adaptability of the team [9]. To be able to address WAD and compare this to the described and planned standard operating procedures, referred to as WAI, one should be able to narrow the difference by adjusting the standard operating procedures to the work that is really done.

The team has better safety management practices in place when these four resilience abilities have taking place prior to (in briefings), during, or following (in debriefings) expected and unexpected circumstances, thereby enhancing the adaptability of the team to operational circumstances and thus improving safety and efficiency. This new Safety II paradigm of RE can therefore enhance MCRM training by changing attitudes and behaviour by training non-technical skills to enhance the adaptability of the team that is part of the socio-technical system during everyday dynamic complex operations. In the following section, the training design of an enhanced CRM training is described.

## 4. Design of CRM Training Enhanced with RE

The objective of enhancing CRM training by introducing RE is to bring the maritime safety to a new level in daily complex and dynamic situations. The difficulty is that resilience research has mainly been focused on academic dialogue applied to theoretical or case studies and to limited trials of specific tools that address a small subset of the overall RE [28]. In this sense, RE must be further operationalised in order to link RE to the MCRM training design.

The enhancement of the CRM training starts with integrating resilience abilities to CRM non-technical skills. As indicated in Section 3, the resilience abilities contain the following resilience cornerstones: anticipation, monitoring, and responding and learning. The CRM non-technical learning objectives for the CRM course consists of allocation, assignment and prioritization of resources, effective communication, assertiveness and leadership, obtaining and maintaining situational awareness, and consideration of team experience including workload management and decision making, as shown in Table 1, where resilience abilities and the non-technical skills are combined to enhance CRM.

**Table 1.** Resilience abilities and non-technical skills enhancing CRM.

| Enhancing Maritime CRM Training by Resilience Engineering | |
| --- | --- |
| Adaptability of the team | Four resilient cornerstones/abilities: Anticipate/Monitor/Response/Learn |
| NTS of Maritime CRM | Effective communication<br>Allocation, assignment, and prioritization of resources<br>Assertiveness and leadership<br>Obtaining and maintaining situational awareness<br>Consideration of team experience including workload management including decision making |

Combining resilience abilities to non-technical skills is a key issue in the design of MCRM training enhancement. Therefore, the resilience abilities form the theoretical lens with which to look at the adaptability of the team on board the vessel [41]. The operationalization of the four resilience abilities is pragmatic and linked with competence, as competence is defined as the knowledge, skills, and personal attitudes applied in various situations [42], both expected and unexpected circumstances within daily operations. Therefore, competence development and training of non-technical skills needs to be focused on changing attitudes and flexible team behaviour in order to enhance the adaptability of the team. Thus, the development and training of competency is needed to increase the ability of students to be a flexible team member, and in doing so to cope, as a team, with operational dynamics and to improve safety performance. Gherardi [43] sees safety as combined competencies that must take place in the context where the work is performed.

Within the CRM training design, the bachelor competencies of the bachelor maritime education and training program provided at the Rotterdam Mainport Institute (RMI) was used as case study for the competencies enhancing the adaptability of the team. From this, the following bachelor competencies can be identified [44], where items two up to eight are relevant for enhancing adaptability and linked to the four resilience abilities of the team.

The following bachelor competencies are defined by the RMI:

1. Research of applied science: the officer is capable of systematically conducting practical-oriented research of applied science that contributes to the solution of a problem within the working environment.
2. Planning: the officer can effectively and efficiently convert a set of goals into a plan.
3. Perform/execution: the officer can independently execute and check the necessary tasks and adjust to the schedule where necessary.
4. Manage emergency/unexpected situations: the officer shows a problem-solving attitude and responds adequately to an emergency.
5. Control unexpected circumstances: the officer takes care of and guarantees the quality of systems and processes and, where necessary, proposes actions of improvement.
6. Communicate: the officer communicates effectively and clearly in both business and social processes within an international context.
7. Manage: the officer provides direction and guidance to the various work processes and the employees involved in order to achieve the set goals.
8. Professionalise: the officer can reflect on his own actions and takes professional ethics into account.

In Table 2, the bachelor competencies and non-technical skills are linked to the resilience abilities of the team. The changing behaviour and attitudes needed for enhancing the adaptability of the team can be observed by behavioural markers. The nautical school in Rotterdam has defined these behavioural markers, but these can still be improved.

**Table 2.** A competence profile of the bachelor maritime training program of the RMI aiming at enhancing adaptability of the team that is part of a socio-technical system in expected and unexpected situations.

| Team Resilience Abilities | Bachelor Competencies of the Maritime Training Program of the RMI | Non-Technical Skills (NTS) |
|---|---|---|
| Anticipate | Planning expected situations | Assertiveness and leadership and consideration of team experience including workload management and other NTS |
| Anticipate, Monitor, Respond | Performing | Obtaining and maintaining situational awareness and other NTS |
| Anticipate, Monitor, Respond | Managing emergency/ unexpected situations | Assertiveness and leadership and consideration of team experiences and other NTS |
| Anticipate, Monitor, Respond | Controlling unexpected situations | Decision making as part of consideration of team experience and other NTS |
| Anticipate, Monitor, Respond | Communication | Effective communication and other NTS |
| Anticipate, Monitor, Respond | Managing | Assertiveness and leadership and other NTS |
| Learn | Reflect on behaviour within expected and unexpected situations, in order to professionalize. | Effective communication and other NTS |

In order to enhance the CRM training design based on RE principles, competence-based training is the key issue and combines both the four resilience abilities of the team with the non-technical skills of the individual bachelor maritime officer contributing to team adaptability, which is part of a socio-technical system in expected and unexpected situations in order to improve safety performance, as can be seen in Table 2.

From our experience, the following remarks need to be made in order to clarify some choices made in the creation of the competence profile in Table 2. Anticipation, knowing what to expect, is communicated in the briefing and continuously repeated along the way by thinking aloud in order to control the operation. Therefore, anticipation is added to every competence and not only to the competence planning. Communication within the team in expected situations can be considered normal and communication in unexpected situations in cases information is needed from an agent in the socio-technical system outside the team in order to adapt, can be considered abnormal. Leadership should build team confidence in expected situations and, in unexpected situations, should remind the team of their adaptability and coach the team members to maintain confidence [45].

In order to acquire the competence profile by changing behaviour and attitudes enhancing the adaptability of the team, the form of the enhanced MCRM training according to Hollnagel [26] needs to be designed by an active process of development, rather than a passive collection of facts and knowledge. From an RE point of view, the form of the enhanced MCRM training design takes the team that is part of a socio-technical system into account, as well as the onboard complex operations taking place in a large variety of conditions. According to Praetorius [35], an increased understanding of how to promote flexibility and adaptability is crucial in the design. Rudolph et al. [46] indicated that the goal of training is to allow trainees to explain, analyse, and synthesise information and emotional states to improve performance in similar situations in the future. Because this is not self-evident, the four-stage learning cycle of Kolb [47], Figure 1, is explicitly applied at the RMI. In this way, student officers learn by doing (having a concrete experience), by thinking about what they are doing (reflective observation), by using lessons learned to modify work practice (abstract conceptualisation), and by applying what is learned (active experiments) [48]. Therefore, the context of the enhanced CRM training design, in our opinion, can be achieved by:

1.  Adding the cognition being distributed in the entire socio-technical system instead of the cognition in the mind of only one trainee [23]. Involving interaction between

different agents so that coordination needs to take place as an attempt to understand the different views of the fellow members and agents in order to update all events that are taking place in the changing environment. Therefore, student officers must be prepared for unexpected events in a socio-technical system, finding ways to enhance adaptability by communicating in abnormal situations and coordinating between different agents that are part of the socio-technical system.

2.  Adding a dynamic/complex context, including unpredictable situations instead of unpredictable static tasks incorporated in the design of the training [49], dynamic involvement can be realised by complexity. The complexity the team is facing creates gaps in everyday operations. This means that the system cannot be adequately described nor controlled as it is currently done in maritime operations. Therefore, the training must involve dynamic, unpredictable situations, which can be realised by including work constraints such as time, information, or available resources, so that the many standard operating procedures (SOPs), to perform the tasks safely and avoid errors, are no longer applicable and need flexibility. Therefore, student officers must be prepared for unexpected complex events, finding ways of applying non-technical skills to enhance adaptability as, according to Wachs [50], the adaptability takes place via a spontaneous process. In order to incorporate complexity without making circumstances too difficult, the recommendations of Röttger et al. [4] must be considered.

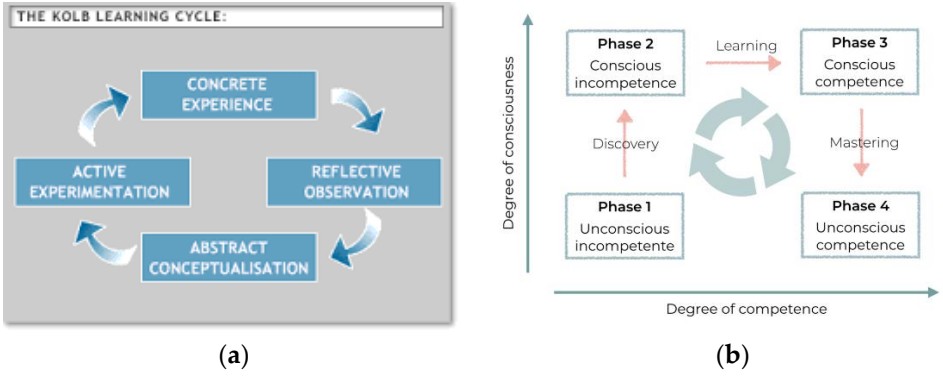

**Figure 1.** (**a**) Learning cycle of Kolb; (**b**) Four stages of competence of Maslow.

When the student officers are having a concrete experience of the RE context, the reflective observation stage of the learning cycle of Kolb (see Figure 1a) should be followed by thinking about what they are doing. This reflective observation phase is crucial for the adaptability of the team [7,26,28] since 'learning' is one of the resilience abilities. Reflective observation is stimulated within the enhanced CRM training by adding ample time for briefing (as salient aspect) and debriefing (as anticipation aspect) to the design of the training [6]. Feedback and feedforward techniques are to be used as a mechanism to control the situation, e.g., to find out why operations were successful and what the student officer will do next time to improve. In case the enhanced MCRM training design is one of concrete experience and reflective observation, the student will become much more conscious of "why operations go right" and "why operations go wrong" [9]. Therefore, a common response such as freezing when witnessing an unexpected event can also be reflected upon in order to discuss how to deal with possible 'freezing' behaviour [51]. In the reflection of the unexpected situation, no specific result is expected because, as Wahlström et al. [52] have asserted, it is difficult to expect a result in the reflection because team resilience arises spontaneously through task-sharing, coordination, and shared understanding.

The form of the enhanced CRM training design will also be added with the recommendations of Röttger et al. [4] for improving effectiveness and ensuring that the design does not become too complex:

1.  In order to be effective, training design should be directed at specific behaviours and the best practices in a given context of application.
2.  In the training design, make sure to train complete teams instead of individual team members.
3.  Determine training needs at the beginning of the training design and focus the training on those non-technical skills and procedures that do not sufficiently comply with the behavioural standards.
4.  Provide opportunities in the training design for repeated training and debriefing of behavioural standards.
5.  Follow a step-by-step approach in training design instead of trying to improve everything at once. Focus on no more than three behavioural standards at a time. If a standard is exhibited repeatedly and sufficiently, go on to the next one.
6.  In simulators training design, technical and non-technical skills should be trained jointly, because they must be jointly executed on the bridge.

The enhanced MCRM training design can be used in different learning environments, e.g., both classroom lectures and simulator-based exercises. According to Praetorius [35], the focus of CRM training is based on simulator-based exercises as, in classroom lectures, the focus is more on the abstract conceptualization of non-technical skills and analysis of incident reports [53]. In addition to simulator-based exercises, room for active experiments in classroom lectures can be created by serious game-based exercises. These context specific trainings, called serious games are developed at the RMI with virtual reality glasses, and the form is based on the enhanced MCRM training design. This provides that the student, as part of a team in a classroom, will be able to experience a context-specific experiment in order to create a concrete experience, and is able to reflect on the adaptability of the team in a dynamic, unpredictable situation. These kind of innovative learning environments create more opportunities for training and developing the resilience abilities and non-technical skills.

As learning takes time, the learning of non-technical skills take even more time [54], so the main challenge of the enhanced MCRM training design should not be limited to once or twice. The maritime education and training program but should be repeated constantly during the 4-year curriculum, as performance improvement of the competencies leading to an improvement in the adaptability of the team that is part of the socio-technical system in ever-changing situations is an ongoing learning process [55].

Success depends not only on effective, enhanced CRM training design, but also on important elements such as the consciousness of the student officers of their competencies and non-technical skills [54]. The student's consciousness of being (in)competent follows the four stages of competence of Maslow (see Figure 1b) [56]. For students to come from unconscious incompetence to conscious (in)competence takes time and is a learning process. Feedback and reflection are needed and are described by Kolb [47] as 'the learning process whereby knowledge is created through the transformation of experience (p. 38)'. Therefore, the design must take the four stages of competence of Maslow into account, in order to change student officers from being unconscious in their learning difficulties to becoming conscious, skilled, and even professional-based. In the four-year curriculum, increased yearly complexity will account for the competence development.

In the following section, the case study of the "Rotterdam Approach" is explained, in which the CRM training design is enhanced with RE, aiming at changing attitudes and behaviours to create sufficient adaptability within teams and with which to improve safety performance.

## 5. Design and Implementation of a More Effective Training Program by Including the Qualities of Resilience Engineering

In this section, the design and implementation of an enhanced maritime CRM training program is described and is part of the bachelor curriculum of the nautical school in

Rotterdam, The Netherlands. This program is based on the recommendations mentioned in previous sections.

The maritime training system in the Netherlands differs from the rest of the world, as student officers are trained to perform both the nautical as well as the engineering duties and are therefore referred to as "maritime officer". A few other countries implemented this integrated training also.

In the Netherlands, this started in 1985 with the aim to decrease the manning costs on Dutch flagged vessels by operating the vessels more efficiently and effectively with a flexible crew, which can balance the workload between both disciplines. Unfortunately, most of the ships were conventionally operated by mono-trained officers, where the maritime officer worked as either a deck officer or an engineer. Thus, it seems that the integrated training program was not very effective. However, it provides many advantages regarding knowledge and understanding of counter-discipline during the normal and abnormal operations, because an understanding of the situation is gained more quickly, and more people can contribute to solve the problem.

In the Netherlands, both a vocational and bachelor programme for Maritime Officer run in parallel to fulfil the STCW requirements for qualification at management level. Apart from the more practical skills, as required by the STCW, the bachelor programme focuses also on the academic education for the development of academic skills, such as research, analytical, and critical thinking skills, to contribute to the required bachelor competencies. These academic skills provide a major contribution to the development of the non-technical skills of the CRM training because they contain similar elements such as communication, the analysing of information, and decision-making, and are trained during project work. During project work, student officers must collaborate, communicate, plan the project, and reflect on learning objectives, which contributes to STCW and research skills. In this way, a strong connection is created between practical and academic skills.

The bachelor program of four years (8 semesters) is composed of three levels of competence development with increasing complexity and includes the STCW requirements at both the operational level and the management level, which are assessed at the end of semester four and six, respectively. This structure of the curriculum has a lot of similarities with the research work of Nazir et al. [57], where it was proposed to start with basic components of the process, advancing to real-time operations, and reaching high technical and relational complexity that needs to be handled in situations with limited time and uncertainty in data. This also means that the three levels of competencies and non-technical skills are required, where each level is described by behavioural markers in order to assess the behaviour.

After finishing these stages of training, the student can be considered sufficiently competent to start working on the ship on tasks that match the level of competence for normal as well as abnormal operations, but recurrent training is required, as mentioned in Section 2.

The overview of the program (see Table 3) shows how the different non-technical skills are scheduled, and the coupled instructional mode/method is based on the design, as mentioned in Section 4. The size of the team increases from two student officers in semester one to up to four in semester eight. However, an individual assessment is required in order to assess the proficiency of mainly technical skills such as the appropriate use of bridge equipment, route planning and monitoring, applying the COLREGS, etc. The total number of hours spent on the training is about 80 h.

**Table 3.** The overview of the program.

| Learning Subjects | Semester | Learning Method | Assessment |
|---|---|---|---|
| Human element | 1 | Lectures | |
| Teamwork | 1 | lectures; project work; practical | report and oral |
| Situation awareness | 1 | lectures; practical; training ship | |
| Effective Communication | 1 | lectures; project work; practical; training ship | report and oral |
| Leadership and managerial skills | 2 | lectures; practicals | |
| Decision making | 2 | lectures; practicals | |
| Standard Operating Procedures (SOP) | 2 | lectures; practical | |
| Automation awareness | 2 | lectures; practical | |
| Rehearsing and integration of NTS; basic knowledge of resilience | 3 | lectures using videos to recognise behaviour. practical with more focus on tasks on bridge and engine room and more complex scenario's project work, analysing incident reports | Assessment bachelor competence level one, including both the technical- and non-technical skills at STCW operational level, using bridge and engine room simulator. |
| Practical half year at sea. | 5 | One of the shipboard training tasks focuses on non-technical skills. | |
| Repetition and deepening the understanding of NTS, integrated with the technical skills at management level. | 6 | lectures; analysing incident reports. practical | Assessment bachelor competence level two, including both the technical- and non-technical skills at STCW management level using bridge and engine room simulator. |
| Enrich the NTS: - conversation techniques - deepening the understanding of the NTS as a manager on board with focus on leadership, workload and managing emergencies. | 6, 8 | lectures; practical; shipboard training task with a survey and interview of ship's crew | Assessment bachelor level three, related to the professional skills |

In reference to Table 3, the lectures focus on the theoretical part of the non-technical skills and consist of a mix of teaching methods, such as explaining the concepts, case studies of maritime accidents and group discussions about the case studies, and their experience during the shipboard training. The practical lessons are performed by using bridge- and engine room simulators, the engine room, and serious gaming, where during the debriefing student officers review their performance on both their technical and non-technical skills.

During the final year of the program, the non-technical skills are expanded with elements that are important for the management on board, including conversation techniques, and workload and managing emergencies. During their second training period at sea, student officers must write a report on how these elements are applied on board, the correspondence with the literature, and a reflection on their own development as a prospective officer. After returning from sea, these elements are also trained and assessed on the simulator by performing scenarios with a team that consists four student officers, each with their own function. The functions used for the bridge team are: "head of watch", "navigator−1", "navigator−2", and "helmsman". The "head of watch" is the leader of the team and is responsible for the overall operation.

Consideration has been given to make a fixed role and tasks division as described by Carnival UK and Princess Cruises [58], because this ties in very well with the resilience aspects. However, students in that case are not asked to think about the division of tasks in the various circumstances. Furthermore, only a small number of student officers will come to sail on ships where this division of roles has been made, because a division of roles is often not possible because the size of the crew is too small.

The scenarios for the bridge team contain the following consecutive phases and elements:
1. Preparation by the "head of watch":

- voyage planning from a crowded anchorage to the berth,
- allocation of tasks to the team members,
- checks and tests of the bridge equipment including the settings and procedures,

- informing engine room and deck team (bosun) about departure from the anchorage.

2. Briefing of the team by the "head of watch":

- sharing information of the voyage plan and current situation of the vessel to create a shared mental model,
- leadership style used and explaining what this style means for the team,
- assigning of tasks and check if tasks are clear for everyone and no tasks are missing,
- encourage the team to use the appropriate team skills such as: "closed loop communication"; "feel free to challenge"; "speak up if there is a deviation from the plan, a loss of situation awareness or any other concern"; "inform if someone feels overloaded with tasks"; "think out aloud of planned actions or intentions",
- during the briefing, the simulation runs and creates a dynamic situation where vessels approach, leave or pass the anchorage.

3. Execution of the voyage plan:

- the "head of watch" monitors the team standing at distance behind the workstations of the two navigators and the helmsman in order to keep a good overview. The main tasks are performed by the two navigators and are cross checked by the head of watch before execution,
- the scenario is as follows: after heaving up anchor, own vessel proceeds to the pilot station. There are several inbound and outbound vessels, which report by VHF to the pilot, VTS. Moreover, inter-ship communication between vessels is made to avoid collision. Disturbances are introduced by internal and external influences, e.g., malfunction of the bridge equipment, internal or external phone calls, or persons entering the bridge. Own ship to deviate from the plan due to an emergency that occurs on own vessel, e.g., cargo fire, cargo leakage, bilge alarm hold, etc. Sharing information of the emergency with all parties concerned, i.e., deck crew, VTS, pilot, vessels in close vicinity, and management of the ship. To manage the emergency, one of the navigators to leave the bridge and assist the emergency team and reassigning of tasks is required. For an emergency related to the cargo, a stowage plan and dangerous goods list is on the bridge. After controlling the emergency, the vessel to return to the anchorage or proceed to the pilot station.

4. Debriefing

- The final phase of the scenario is the debriefing, which starts with a brief overview of the team reporting the positive and negative results and the reasons why they were successful and what they will do the next time to improve. Next, a comprehensive debriefing is made using the observations from the assessors.

In order to gain an understanding of how other training institutions around the world executed their implementation of the Manila amendments to the STCW in their curricula, a questionnaire was sent to 23 lecturers who are familiar with the training of the required skills of "resource management, leadership and teamwork".

The questions focused on the goals of the training, frequency, monitoring of the change of behaviour, and the joint training of technical and non-technical skills on simulators and how they have implemented principles of RE.

The results of the responses ($n$ = 15) were:

- Goals of the course are improving safety; efficiency; preventing accidents by human factor; working as a team.
- Number of courses varies between two and five with a total duration of 40 h on average. Two institutions mentioned that the subjects are fully integrated in all practical and simulator lessons.
- Courses are scheduled in years 2, 3, and 4, but mainly in years 3 and 4
- Teaching method used is mainly a combination of classroom lectures, case studies, and simulator-based training.
- When using the simulator for training of the non-technical skills, the focus is on both the technical skills and the non-technical skills, and both are assessed or evaluated

- Assessment of the skills is by simulator or a combination of simulator with written exam.
- During the courses, the change of behaviour is mainly monitored by using observation forms and behavioural markers.
- The individual and team of officers on the bridge or in the engine room are trained.
- The implementation of principles of RE is different during briefings and debriefings; only 66,7% pay attention to 'why things go right' and 'why things go wrong', only 53,3% pay attention to monitoring aspects and information important for the job, only 66,7% pay attention to responding, e.g., discussing SOPs, communication, control, and coordination taking place in the socio-technical system, and only 56,3% train last-minute risk analysis in unexpected events.

## 6. Summary and Discussion

The results of the survey, demonstrates that most maritime training institutes implemented the training courses in line with the recommendations by Röttger [4] and that the goal is to prevent accidents caused by humans. However, less attention is paid to the fact that the percentage of things that go right are much higher compared to things that go wrong. This is because, people tend to pay much more attention to things that go wrong than to things that go wright [59]. However, it makes sense to try to understand why daily operations go right and go wrong. The goal of RE is to increase the number of day-to-day operations that go right rather than to reduce the number of things that go wrong, because the latter will be a consequence of the former, as discussed in Sections 3 and 4.

Moreover, only a part of the socio-technical system is being trained, because only the student officers who are at school follow the training program. Moreover, several elements cannot be sufficiently trained, such as cultural differences and assertiveness in a homogeneous group of student officers in terms of culture, age, and rank.

The IMO model course [53], which can be regarded as standard, is susceptible to different interpretations. This could be remedied if maritime institutions take the opportunity to learn from each other by exchanging their experiences, for instance in the process of the development of behavioural markers, creating realistic simulator scenarios, and having a good understanding and integration of everyday work, so that they can enhance the adaptability of the team that is part of a socio-technical system in order to improve safety. Even though several shipping companies send their officers for recurrent training, it seems logical to include that this part becomes mandatory in addition to the other safety training courses, where the training is conducted with the ship's team. In doing so, cultural awareness, assertiveness, and team resilience are further developed.

The experiences gained with the enhanced Rotterdam program as described in Section 5 looks promising, but following points require further attention:

- A set of behavioural markers is used for assessing the (change of) behaviour, but some of these are still open for multiple explanations [60]. The intention is to consult other training institutes and the professional field to improve the process of the development of a more uniform set of behavioural markers and associated scenarios which cover both the normal (everyday work) and the emergency situations.
- The resilience cornerstones are used for the RE theoretical lens, but the abilities are still open for multiple interpretations, i.e., anticipation, according to Turan et al. [34], considers a longer time-horizon beyond the current operation, applying a broader perspective in which each team member should have a clear definition of their role, tasks, and responsibility in order to ensure operational performance. The intention is to do further exploration on board the vessel to understand the resilience abilities better in real life settings.
- During the assessments, the focus is mainly on the competence of the individual student, which is required for certification. The assessor takes notes on the contribution to the team, but team performance is not assessed yet. The authors observed a bridge team making use of the competence profile in Table 2, which showed that, in a

case where the team is, for example, not proactive and does not anticipate the next condition of the adaptability of the team, it is not improving along the assessment. At this moment, we are still discussing how to incorporate the team performance as part of the assessment. On board it is easier to assess the adaptability of the team, looking for team resilience as the crew itself is an entity; therefore, we agree with Praetorius et al. [61] that "research to the resilience engineering cornerstones on board can help to gain insights and generate vital knowledge on team dynamics based on respondents' previous experiences."

- During the debriefing of the scenario, the discussion focusses mainly on what went wrong. Student officers find it hard to answer the question "why things/operations went right". The four stages of Maslow, becoming conscious of the (in)competence of the non-technical skills, the bachelor competencies, and the adaptability of the team, will help to answer "why things/operations went right". It also gives student officers an opportunity to observe each other's actions and reflect on their practice with peers [48]. Discussing "why things went right" also improves everyday life and creates confidence, according to the positive psychology scientific study [61]. This behaviour will also help on board once the student officers have become professionals, as the young professional is used to giving feedback and reflecting, and is able to learn. In the past, leaders on board tended to receive little feedback on their behaviour, and followers tended not to question or break their governing norms [62].

- Student officers who performed their practical training on board showed that they rely more and more on standard operating procedures (SOPs) to solve problems. These SOPs can also pose a troubleshooting hazard for problems on board that cannot be covered by procedures [63]. Recently, serious gaming has been introduced in the program to improve the competencies that add up to resilience in the face of unexpected and emergency situations, which provide student officers with the skills they need to handle situations that go beyond what can be expected in classroom lectures [64].

- The training scenarios containing emergencies contribute to the development of appropriate behaviours during emergencies, where the team uses the phases of orientation, evaluation, and decision making to think their way through a novel situation. However, it has been observed that most student officers failed to respond in an appropriate way and in time due to a lack of pre-planned behaviours and appeared to "freeze" [51]. This was most frequently observed with the student with the function "head of watch", who was responsible for the operation. If the other team members are familiar with this behaviour, they can recognise this and resolve this freezing by challenging the "head of watch". The intention is to introduce the emergency scenarios in an earlier stage in the training program in order to enrich the pre-planned behaviours and to overcome "freezing".

- For the measurement of the effectiveness of the training program it is important to understand what the effect is on the improvement of safety. Starting next year, we want to start to implement this at the nautical school in Rotterdam. The Kirkpatrick levels of training evaluation [65] are often used to assess the effectiveness of the training. The four levels of evaluation are:

1. the reaction of the student and their thoughts about the training experience,
2. the students' resulting learning and increase in knowledge from the training experience,
3. the students' behavioural change and improvement after applying the skills on the job,
4. the results or effects that the students' performance has on the business.

- A maritime training institute can only assess the behavioural change to level 2 and further research is required on how to assess the transfer from the simulated environment to the real world and how this affects the performance on board.

- When more attention on board the vessel is paid to non-technical skills and the adaptability of the team, the scenarios and NTS needed in the enhanced MCRM

training will be better defined so that what is learned in the MET becomes more relevant for work on board [60].

As a final note, the motivation to write this paper has been to provide suggestions on how to enhance MCRM training. We are curious about your thoughts and would like to invite everyone to share their experiences, so as to learn from each and cooperate on improving the effectiveness of the MCRM training, since we regard this as an ongoing learning process.

**Author Contributions:** Background and history on MCRM, questionnaire design, and scenario description, J.G. Background on RE theory and connection with effective training program, M.v.d.D. First draft and description of the 'Rotterdam' approach, J.G. and M.v.d.D. Supervision, review and editing, H.v.d.B. All authors have read and agreed to the published version of the manuscript.

**Funding:** This research was funded by the Research Centre for Sustainable Port Cities, Heijplaatstraat 23, 3089 JB Rotterdam, The Netherlands.

**Institutional Review Board Statement:** Not applicable.

**Informed Consent Statement:** By answering and returning the questionnaire, the respondents did give informed consent.

**Data Availability Statement:** The completed questionnaire and the anonymous answers are available on request from the corresponding author.

**Conflicts of Interest:** The authors reported no potential conflict of interest.

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
