# Peer review of "Enhancing Maritime Crew Resource Management Training by Applying Resilience Engineering: A Case Study of the Bachelor Maritime Officer Training Programme in Rotterdam"

_education, doi:10.3390/educsci11080378_

Round 1

Reviewer 1 Report

Dear Authors, 

It is a very interesting idea. However, The paper has to be extensively improved for reviewing and to be published in this high-quality journal. Unfortunately, there is neither clear methodology nor results presented which can prove that Enhancing Maritime Crew Resource Management training by applying Resilience Engineering could improve maritime safety. Methodology, analysis and results are not clearly presented. For instance, there is no information on the countries that participated, instructors, questions, statistical method, and appropriate discussion.

You will find underlined comments further in the text.

Hope this review will be helpful in improving your paper quality.

Best regards,

Reviewer.

Reviewer 2 Report

 The study appears to be aimed at improving CRM training for maritime officers. CRM training is conducted according to international standards in accordance with the regulations of IMO. This study introduces the Rotterdam curriculum as an example.

Questions about this study are:

1. CRM (CRM, including BRM, ERM, BRTM, etc.) is a training that has been conducted for a long time. Since 80% of accidents are caused by human error, please introduce whether there is any evidence that developing this training course contributes to safety.
2. The readability is little poor because there is a lot of content in the body. In particular, it is more difficult because the procedures of this study are not described at the beginning of the text.
2. The text contains many things that are not necessary for this study. Please describe the content that is improved according to this research method with emphasis.
4. Figures 1 and 2 seem more natural to transform into tables.

In spite of the above major question, this paper is considered suitable for the scope of the journal Education sciences.

Reviewer 3 Report

The topic is interesting and article is valuable by means of its content and appropriate scientific style. The provided background to place the problem of research is adequate and complete. There are some remarks:

1. As one of the drawbacks is absence of any graphical data in the 1-3 sections. It makes the paper a bit boring for reader. I suggest to add some visual data here and it could be make paper stronger. Maybe some bullet lists could be changed to diagrams, sketches etc. The same about bachelor competencies in chapter 4.

2. Figure 2 indeed is a table. Also other tables used in the paper contains important information but they are not exciting. As a suggestion, consider a bit wider colour palette. Actually, I suggest to adhere more to some functional diagrams and sketches representation instead of very similar tables, bullets/numbers lists over the paper.

3. The last but very important thing is some lack of scientific novelty in the presented research in its current form. At least, is it not shown and emphasized clearly enough. It was sometimes rather hard to distinguish the authors' ideas from existing in the literature. Therefore I suggest carefully check and point out the main contributions made in this work.

Round 2

Reviewer 1 Report

The Authors improved the quality of the paper. 

Best regards, 

Reviewer 

Reviewer 2 Report

I sincerely responded to previous comments, and the author's thoughts were well revised and reflected in the text.

As a research paper on education and training, it is judged that it can be published as a paper that proposes a just improvement.

Reviewer 3 Report

Those changes made to the revised version have improved the quality of the manuscript.